# Optimizing GC-IMS for Pork Volatile Fingerprinting: Effects of Incubation Conditions and Medium on Aroma Profiles

**DOI:** 10.3390/foods14234164

**Published:** 2025-12-04

**Authors:** Lei Yu, Binbin Wang, Ziwei Xu, Kaili Ge, Yihan Yuan, Xiangbin Ding, Xiaoming Men, Keke Qi

**Affiliations:** 1Tianjin Key Laboratory of Agricultural Animal Breeding and Healthy Husbandry, College of Animal Science and Veterinary Medicine, Tianjin Agricultural University, Tianjin 300392, China; yl653683@163.com (L.Y.);; 2Institute of Animal Husbandry and Veterinary Science, Zhejiang Academy of Agricultural Sciences, Hangzhou 310021, China; bbwzaas@126.com (B.W.); zjsnkyxzw@163.com (Z.X.)

**Keywords:** pork, VOCs, GC-IMS, incubation conditions, medium environments

## Abstract

Volatile organic compounds (VOCs) are key aroma determinants in pork, and gas chromatography-ion mobility spectrometry (GC-IMS) is an effective technique for their detection. However, the detection conditions for pork using GC-IMS have yet to be standardized. This study employed GC-IMS to investigate the effects of incubation (temperature/duration) and medium (water and different concentrations of NaCl) environments on VOCs in pork. Statistical analyses including t-tests, PLS-DA, and OPLS-DA were employed to assess VOC differences. The results showed that: (1) VOC diversity and intensity increased with incubation temperature and time, with optimal signals obtained at 100 °C for 20 min. (2) The sample medium significantly influenced aroma release. When the medium contained 10% NaCl, the relative content of aldehydes increased, showing that these compounds were optimally released. (3) Under the optimized conditions, 15 key differential VOCs were identified from different muscle tissues, including 10 aldehydes, 2 esters, 1 ketone, 1 alcohol and 1 ether. This work establishes practical GC-IMS parameters for pork VOC analysis and provides a reliable reference for flavor-related studies.

## 1. Introduction

Pork is one of the most widely consumed meats globally due to its high yield, rich nutritional value, and unique flavor. With the growing consumer demand for food quality, pork flavor has become a key factor affecting purchasing intentions [1]. Pork flavor comprises taste and aroma, where aroma mainly arises from volatile organic compounds (VOCs) (e.g., lipids, amino acids, etc.) in meat. These precursors are converted into VOCs through lipid oxidation, Maillard reaction, and amino acid degradation, and the types and contents of these VOCs directly determine the flavor quality of pork [2]. However, VOCs are characterized by diversity, low concentration, and poor stability, necessitating a stable and reliable detection method.

Current detection methods include electronic noses (E-nose), gas chromatography-mass spectrometry (GC-MS), proton-transfer-reaction mass spectrometry (PTR-MS), Gas Chromatography-Ion Mobility Spectrometry (GC-IMS) and others [3]. Among these, GC-IMS stands out as a powerful technique for VOC analysis, featuring ultra-trace detection capability, precise discrimination of isomers, direct sample analysis, and intuitive visualization of results [4]. Specifically, GC effectively separates VOCs in complex matrices, while IMS further differentiates compounds based on differences in ion migration times, enabling the detection of trace substances at ppbv (parts per billion by volume) levels and even pptv (parts per trillion by volume) for certain compounds [3,5]. This technique has been widely applicable in non-targeted trace VOCs detection currently, as well as quality control in fields such as food safety and environmental monitoring [5,6].

In GC-IMS analysis, key influencing factors include incubation temperature, incubation time, and sample media. Previous studies have established that incubation temperature significantly affects the types and contents of VOCs, mainly through regulating protein-flavor substance interactions [7,8]. For example, elevated temperatures enhance the volatilization of short-chain aldehydes, low-molecular-weight ketones, and short-chain fatty acids, while excessive causes thermal decomposition of unsaturated aldehydes and ethers [9,10]. Incubation time is closely related to VOCs generation, insufficient incubation time reduced VOCs production due to competition between thermal reactions and lipid oxidation reactions [11], while prolonged incubation leads to the accumulation of aldehydes and alcohols [12], excessively long incubation can degrade heat-sensitive compounds (e.g., sulfur-containing compounds) [13] and triggers secondary reactions (e.g., conversion of aldehydes to acids) [12]. As a common aqueous medium, water influences the solubility and distribution equilibrium of flavor substances and regulates protein folding (e.g., exposure of hydrophobic regions) to alter VOCs binding capacity [7], thereby playing a crucial role in flavor release and retention. Brine concentration affects liquid–gas equilibrium in food: the salting-out effect reduces solute solubility in the aqueous phase, promoting the volatilization of carbonyl compounds (e.g., aldehydes, ketones) and short-chain fatty acids [14]; meanwhile, changes in sodium ion concentration affect protein degradation kinetics, indirectly regulating VOCs formation [7,15].

GC-IMS exhibits high sensitivity to variations in sample preparation and detection conditions. At present, no standardized protocol has been established for analyzing pork VOCs using this technique [16]. Despite the increasing use of GC-IMS for meat analysis, no consensus exists on standardised detection parameters. While GC-IMS has been applied with established temperature and time parameters in studies on olive oil [17], the role of the sample medium has not been fully examined. In comparison, GC-MS methods are relatively well-developed, with systematically evaluated parameters such as incubation temperatures (30–100 °C) and durations (10–50 min), which have been established across various food matrices including beef, pork, and dairy products [11,18,19].

To address this gap in pork research, this study employed several statistical analyses to compare VOC profiles under different experimental conditions. An optimized detection protocol was established by quantitatively evaluating key influencing factors, and aroma-active compounds were identified based on the relative odor activity value (ROAV). The formation pathways of flavor compounds and critical influencing variables were elucidated, providing a methodological reference for flavoromics research in pork.

## 2. Materials and Methods

### 2.1. Materials

Muscle samples for optimizing experimental conditions in this study were obtained from the Lvjiahei (LJH) pig, which was bred by the Zhejiang Academy of Agricultural Sciences. This breed is known for its tender texture, high intramuscular fat (IMF) content [20], and superior flavor. LJH pigs originated from Yangdu Technology Ranch. Experimental pigs were housed indoors, with ad libitum access to a corn-soybean meal-based diet and water. All slaughter procedures were conducted by Qinglian Food Co., Ltd. in strict compliance with the Chinese National Standard (GB/T 17236-2019) [21] and relevant animal welfare and ethical guidelines. After slaughter, carcasses were refrigerated at 4 °C, and the core temperature of the thigh muscle was ensured to reach 4 °C within 24 h postmortem. To ensure methodological consistency, experimental conditions were optimised using samples of the *longissimus dorsi* (LD) muscle taken from the third to terminal ribs on the left side of the same pig. In addition, to validate the feasibility of the study, muscle samples were collected from five additional pigs subjected to the same slaughter procedure. All sampling procedures were conducted after 24 h postmortem. Muscle tissue was subjected to mechanical disruption using a high-speed tissue homogenizer (Model MEAT-3, Tenovo International Co., Limited, Beijing, China) at 15,000 rpm for 10 s. The resulting homogenate was aliquoted into cryovials and stored long-term at −80 °C.

### 2.2. Experimental Design

3 g homogenized pork was accurately weighed to 0.001 g using an analytical balance. The weighed sample was immediately placed into a 20 mL headspace vial (Hanon Group, Shangdong, China). The experiment was divided into two parts to identify the optimal GC-IMS conditions for pork: incubation condition optimization and medium condition optimization. (1) Incubation condition optimization: Two key variables (incubation temperature and time) that significantly influence GC-IMS results were evaluated. Three temperature gradients (60, 80, and 100 °C) and two time gradients (15 and 20 min) were set to test the optimal combination, with selection based on the GC-IMS detection results. For each combination of temperature and time, three independent replicate samples were prepared and each was analyzed once by GC-IMS to ensure reproducibility. (2) Medium condition optimization: After the determination of the optimal temperature and time, five media were evaluated to identify the optimal medium: air, pure water, 10% NaCl solutions, 18% NaCl solutions, and saturated NaCl solutions (26%, *w*/*v*). Similarly, each medium condition was tested in triplicate, with one GC-IMS analysis performed per replicate sample, to assess consistency across repetitions. (3) To validate the feasibility of the optimized conditions, two distinct muscle types, namely LD and serratus ventralis (SV), from 5 LJH pigs (n = 5), were selected for verification.

### 2.3. Gas Chromatography-Ion Mobility Spectrometry

For the precise calculation of the retention indices (RI) of VOCs, six methyl ketones were used as reference standards for retention index calculation: 2-butanone, 2-pentanone, 2-hexanone, 2-heptanone, 2-octanone, and 2-nonanone (Aladdin Biochemical Technology Co., Ltd., Shanghai, China). The GC-IMS analysis was performed on a FlavourSpec^®^ instrument (G.A.S. mbH, Dortmund, Germany). GC conditions: shaking at 500 rpm; a needle temperature of 85 °C (set according to instrument recommendations); a headspace injection volume of 500 μL; analysis with non-diversion mode. Chromatographic conditions were as follows: initial flow rate of 2.00 mL/min maintained for 2 min, linearly increased to 10.00 mL/min over 8 min, and then linearly increased to 100.00 mL/min over 20 min, with 100.00 mL/min maintained for 10 min; chromatographic run time: 30 min; a column (MXT-WAX, 30 m × 0.53 mm × 1.0 μm, Restek Corporation, Bellefonte, PA, USA) temperature at 60 °C. IMS conditions: ionization source: the ionization source was a tritium (^3^H) source with an activity ranging from 75 to 370 MBq, which is below the 1 GBq threshold for an exempt radioactive source; drift tube length of 53 mm; drift tube voltage at 2700 V; drift tube temperature at 45 °C; drift gas: high-purity nitrogen gas (≥99.999%, JinGong Gas Co., Ltd., Qingdao, China); flow rate of 75 mL/min; positive ion mode.

### 2.4. Statistical Analysis

GC-IMS experimental data of samples were analyzed using VOCal software (v04.12 364; G.A.S. mbH, Dortmund, Germany), which was employed for analytical processing and graphical visualization. Color representation of compound abundance in the fingerprint profile: red (high), white (low), and black (not detected). A mixed standard of six ketones was analyzed to establish calibration curves (R^2^ = 0.996) for retention time (RT) and retention index (RI). Subsequently, the RI of target VOCs was calculated based on their RT as follows.$$y=664.39x^{2}-2715.7x+3594$$
In this equation, *y*: the retention index (concentration range: 800–1400), *x*: log [retention time/second] (concentration range: 2–3).

Qualitative analysis of target VOCs was performed by searching and comparing against VOCal’s built in database, combining the GC retention index database (NIST, 2024) and IMS drift time database. Vocal’s data processing plugins, including Reporter and Gallery Plot, were utilized to generate two-dimensional spectra, fingerprint plots, and odor characteristic markers for comparative analysis of VOCs between samples.

The volatilomics dataset underwent statistical analysis via the MetaboAnalyst online platform (https://www.metaboanalyst.ca/ accessed on 22 October 2025), employing a *t*-test, partial least squares discriminant analysis (PLS-DA), and orthogonal partial least squares discriminant analysis (OPLS-DA) [22]. Significant differential VOCs were identified based on a variable importance in projection (VIP) score > 1 and *p* value < 0.05. Results of GC-IMS analysis and their relative odor activity value (ROAV) were presented through multimodal visualization. These visualizations were generated using ChiPlot (https://www.chiplot.online/, accessed on 22 October 2025). The ROAV was used to evaluate the contribution of individual VOCs to the flavor profile. The ROAV of compound A was calculated as follows [23].$${ROAV}_{A}=\frac{C_{A}}{T_{A}}\times \frac{T_{M}}{C_{M}}\times 100$$
*ROAV_A_*: relative odor activity value of target compound; *C_A_*: relative content of target compound; *T_A_*: threshold of target compound; *C_M_*: relative content of greater contribution compound; *T_M_*: threshold of greater contribution compound. The odor threshold values used for ROAV calculation were sourced from the reference book [24].

## 3. Results and Discussion

### 3.1. Effect of Incubation Conditions on Vocs

Total GC-IMS detection results are presented as spectral fingerprints in Figure 1A. In the experimental investigation on the effects of thermal processing parameters (incubation temperature and time) on VOCs, 76 distinct compounds were identified. These included 24 aldehydes, 15 alcohols, 10 ketones, 9 esters, 7 heterocyclic compounds, 3 ethers, 1 carboxylic acid and 7 unknown compounds (Figure 1A). The plot demonstrates the substantial influence of incubation temperature on the VOC profiles. Compared to 100 °C, signal intensities of most VOCs were generally lower at 60 °C and 80 °C. However, certain compounds (e.g., acetone, ethanol, 2-pentanone, and acetic acid) still exhibited detectable signals under these lower incubation temperatures. This plot also revealed the temperature-dependent degradation of specific compounds (e.g., propan-2-ol), which is likely attributable to thermal instability at higher temperatures. Substances with signals lower than the background noise were considered absent, and this was visualized using an Upset diagram (Figure 1B). As shown in the diagram, the number of volatile substances increased with elevation of temperature and extension incubation time. Consistent with the results in Figure 1A, the number of detected compounds at 100 °C was significantly higher than that at 60 °C and 80 °C, and reached the maximum at 100 °C for 20 min. As observed in Figure 1C, acidic substances and aldehydes exhibited relatively large changes in their relative contents. With changes in incubation conditions, the relative proportion of acidic substances decreased from 64.16% to 30.18%, while the relative proportion of other substances increased. Most notably, the proportion of aldehydes increased from 18.93% to 52.81%. This is because in this experiment, only acetic acid was detected among the acidic substances, and its amount was relatively stable; thus, its relative content decreased as the contents of other substances increased. This variation pattern of aldehydes is consistent with other studies, in which the amount of aldehydes reached the maximum at the highest preheating temperature [19]. Aldehydes constituted the most abundant class of VOCs, with their relative abundance positively correlated with increasing incubation temperature and duration. Aldehydes play a critical role in shaping pork’s characteristic flavor due to their low odor thresholds and high concentrations [25]. Alcohols and ketones formed the second most predominant group, sharing fatty acid-derived precursors (e.g., free fatty acids generated from lipid degradation) [26] and common biosynthetic pathways involving fatty acid oxidation or amino acid conversion [27]. Alcohols predominantly contribute floral and fruity aromatic notes [28], while ketones synergistically enhance fruity nuances with aldehydes [29]. Esters, synthesized via alcohol-acid condensation reactions, exhibit ultra-low perception thresholds and can effectively mask undesirable rancid odors, making them sensorially favorable [30]. Heterocyclic compounds, though present in trace amounts, enriched aroma complexity by intensifying roasted flavor attributes [30]. Among acidic constituents, only acetic acid was detected across experimental conditions. Despite its stable quantification, its high odor threshold results in minimized direct olfactory impact.

PLS-DA, a supervised multivariate statistical method, enables sample differentiation. PLS-DA modeling was performed on GC-IMS data from six thermal processing groups (Figure 1D), with cross-validation parameters (R^2^ = 0.914, Q^2^ = 0.814), which indicates robust predictive capability without overfitting. As shown in the PLS-DA plot, incubation time was not a key driver in the production of volatile substances at lower temperatures. At 60 °C and 80 °C, scatter points corresponding to the two incubation times (15 min vs. 20 min) exhibited no distinct separation. In contrast, when incubated at 100 °C, data points for the two time points showed a clear separation. These observations suggested that the VOCs release rate detected by GC-IMS is highly temperature-sensitive, which aligns with the Arrhenius equation [31]. Elevated temperatures increase the migration rates of VOCs from the food matrix into the headspace [7,8], driven by three primary mechanisms: (1) protein denaturation reduces VOC-binding capacities, particularly enhancing the release of hydrophobic aldehydes [7]; (2) thermal lipid oxidation generates more aldehydes and ketones; (3) the Maillard reactions between reducing sugars and amino acids produce heterocyclic compounds (e.g., furans, pyrazines) [19]. At low temperatures (60 °C and 80 °C), fewer VOCs are released under the same incubation duration, a phenomenon associated with saturated vapor pressure. Specifically, due to the inherently low saturated vapor pressure at low temperatures, even with prolonged incubation, the increase in the concentration of VOCs in the headspace remains negligible [32]. Additionally, low temperatures require longer equilibrium times; the kinetic process of volatile substances diffusing from the sample matrix to the headspace proceeds slowly at low temperatures, making it challenging to achieve release equilibrium within standard incubation durations [33].

Based on the comprehensive analysis and discussion above, the optimal incubation conditions for GC-IMS analysis of pork VOCs are incubation at 100 °C for 20 min.

### 3.2. Effect of Media on VOCs

Water can either promote or inhibit specific reactions—including hydrolysis, Maillard reactions, and oxidation reactions—thereby influencing the formation of VOCs [2]. While brine (NaCl solution) can adjust the ionic strength of the aqueous phase, thereby influencing the vapor-liquid equilibrium of VOCs [34]. Thus, after determining the aforementioned incubation conditions (100 °C for 20 min), an experiment was conducted to investigate the effects of different media (air, water-addition, brine-addition) on pork VOC profile.

To visually compare the effects of media on VOCs, the peak intensity signals generated by GC-IMS were represented as spectral fingerprints (Figure 2A). As shown in this figure, the contents of octanal, nonanal, etc., decreased after water addition but increased after brine addition. The reason is that a high-moisture environment may reduce oxygen solubility, inhibit lipid oxidation reactions, and thereby decrease the formation of certain short-chain aldehyde and others [2,35]. Specifically, 2-methylpentanal exhibited extremely low signal intensity when air was used as the medium, but its signal intensity significantly increased after changing the medium. This may be attributed to the sample hydration, which solubilizes sugar-amino acid precursors and enhances aldehyde generation [36]. Peak areas of each VOC were integrated to calculate their relative contents (Figure 2B). Comparison of relative contents among different groups revealed that when the added brine concentration was 10%, aldehydes, which accounted for the largest proportion, reached a maximum relative content of 68.91%. This is similar to the study by Zhang et al., in which the aldehyde content reached the maximum when marinating duck meat at a 12% salt concentration [37]. Similarly, in a study on mackerel, a 3% salt condition was found suboptimal for the release of key flavor compounds—including benzaldehyde, heptanal (M, D), and (Z)-4-heptenal—with the highest release achieved at 9% salt [38]. Another study on chicken soup also reported that flavor compounds increased initially and then decreased with rising salt concentration, peaking at 2.5% [39]. These variations across studies may stem from differences in the food matrices examined. Thus, it can be inferred that a moderate salt concentration generally promotes the release of flavor compounds, particularly aldehydes.

To further compare the VOC profiles across different media, PLS-DA was performed (Figure 2C), and the top fifteen VIP values were obtained (Figure 2D) to identify potential key compounds. The PLS-DA model had predictive parameters of R^2^ = 0.971 and Q^2^ = 0.836, with no overfitting and good explanatory power. Figure 2C reveals a clear separation among the scatter points of the five experimental groups, indicating a pronounced media effect on VOC composition. In contrast, among the various media, when brine was used as the medium, changes in its concentration had a relatively small effect on VOCs. Additionally, aldehydes and alcohols accounted for the highest proportion among the top 15 potential key substances with VIP > 1. This is because small-molecule aldehydes and alcohols are more likely to volatilize in air but dissolve readily in polar water [40]. Consequently, their content was relatively reduced in the water-added group (0 group), a trend consistent with the observations in Figure 2A. In contrast, the addition of brine resulted in an increase in the overall relative content of aldehydes (Figure 2B), which may be attributed to protein-flavor interactions. These interactions primarily involve physical binding and chemical reactions, with hydrophobic interactions playing a dominant role [7]. NaCl can alter the surface charge of proteins, thereby promoting such interactions and subsequently influencing the release of flavor compounds [41].

The above analysis collectively suggests that alterations in the medium significantly alter the VOC profile of pork. Using 10% NaCl brine as the medium was found to favorably promote the formation and release of aldehydes. This finding is particularly significant, as aldehydes are recognized as critical contributors to the overall flavor profile of pork due to their characteristically low odor thresholds and high abundance [25]; therefore, their modulation can substantially influence the overall sensory characteristics.

### 3.3. Gc-Ims Analysis of VOCs in Pork

Under the established conditions, pork samples from LD muscle (LD group) and SV muscle (SV group) were analyzed: 5 mL of 10% NaCl solution was added to each sample, then the mixture was incubated at 100 °C for 20 min prior to detection. Significant differences in fiber composition exist between these muscles, SV is rich in type I and IIa fibers, while LD is predominantly composed of type IIb fibers [42]. The differences in fiber composition further influence IMF content and the distribution of flavor precursors [42,43,44]. As evidenced by research in Duroc × (Landrace × Large White) pigs, the SV muscle exhibits a 3.25-fold higher IMF content compared to the LD muscle [45]. Previous research has indicated that SV received significantly higher ratings than LD in “meaty intensity,” “umami,” and “fatty flavor.” [46,47] This discrepancy can be attributed to their distinct intrinsic characteristics. Therefore, comparing the VOC profiles of these two muscles under the optimized conditions serves to demonstrate the feasibility of the established GC-IMS method.

In Figure 3A, under the current conditions, the relative content of VOCs in the LD group exhibits a significant difference from that in the SV group. In the OPLS-DA score plot (Figure 3B), the scatter points of the two groups are clustered into two distinct clusters, with a clear separation, indicating obvious differences between the two groups. The OPLS-DA model generated predictive parameters of R^2^X = 0.758, R^2^Y = 0.989, and Q^2^ = 0.987, confirming the model’s reliability in explaining the data. Figure 3C highlights the potential key compounds with the highest VIP scores, the main classes are aldehydes, alcohols, and esters, as well as a small number of ketones and heterocyclic compounds. Among these, substances such as (E)-2-Heptenal-D and (E)-2-Hexenal-D have been identified in black pork soup [48].

To further verify the differences, VOCs were screened based on the criteria of VIP > 1, *p* value < 0.05, and ROAV > 1. A total of 15 key differential flavor compounds were identified across all samples (Table 1). These compounds included 10 aldehydes, 2 esters, 1 ketone, 1 alcohol and 1 ether, indicating that aldehydes are the primary contributors to the flavor differences between LD and SV muscles. Further analysis of ROAVs revealed that most aldehydes contributed more to flavor in the LD group than in the SV group. Existing studies [42] have reported higher aldehyde contents in LD compared to SV, which is largely consistent with the present findings. Among these aldehydes, pentanal, hexanal, heptanal, and octanal are classified as fatty aldehydes. Their formation occurs via the oxidation of unsaturated fatty acids, imparting green and fatty odors [49]. Hexanal, in particular, is regarded as a key compound in pork flavor studies and is often considered a core indicator of lipid oxidation [50]. Octanal is characterized by a complex flavor profile, exhibiting a distinct citrus-like odor [51]. Aldehydes, as a class, are known for their low odor thresholds and play a critical role in shaping the overall aroma of pork [35,48]. Additionally, Table 1 shows that the contribution of 1-octen-3-one was significantly higher in the SV group. This compound is commonly associated with a mushroom-like aroma [52]. Furthermore, it possesses an extremely low odor threshold, allowing it to exert a substantial influence on the sensory profile even at minimal concentrations [53]. This can be attributed to the higher fat content in SV, as ketones such as 1-octen-3-one are primarily generated through lipid oxidation [54]. Specifically, 1-octen-3-one is generated from the enzymatic breakdown of linoleic and arachidonic acids [52].

The GC-IMS experiment conducted under the established conditions can effectively identify the volatile substances in different pork samples, and further determine the differences in the types and contents of VOCs in pork.

## 4. Conclusions

This study investigated the effects of incubation conditions and medium conditions on VOCs in pork using GC-IMS. The experimental results indicated that volatile organic compounds (VOCs) were highly sensitive to temperature variations, and a significant increase in signal intensity was observed when the temperature was raised to 100 °C. In contrast, the influence of incubation time was less pronounced, although prolonging the incubation time to 20 min still resulted in an enhanced abundance of VOCs. Alterations in the medium were shown to modify VOC composition, with the highest aldehyde abundance achieved when 10% NaCl was used as the medium. Consequently, the optimal conditions for pork VOC analysis by GC–IMS were determined to be incubation at 100 °C for 20 min with the addition of 10% NaCl solution. Under these conditions, good reproducibility was observed, and flavor differences among various samples could be effectively identified. This work defines the first systematic optimization of GC-IMS parameters for pork aroma analysis, thereby providing methodological basis and theoretical support for flavoromics research in meat quality evaluation. For future studies, more detailed gradients and additional influencing factors could be established based on this research to further explore the potential of GC-IMS in meat flavor analysis.

## Figures and Tables

**Figure 1 foods-14-04164-f001:**
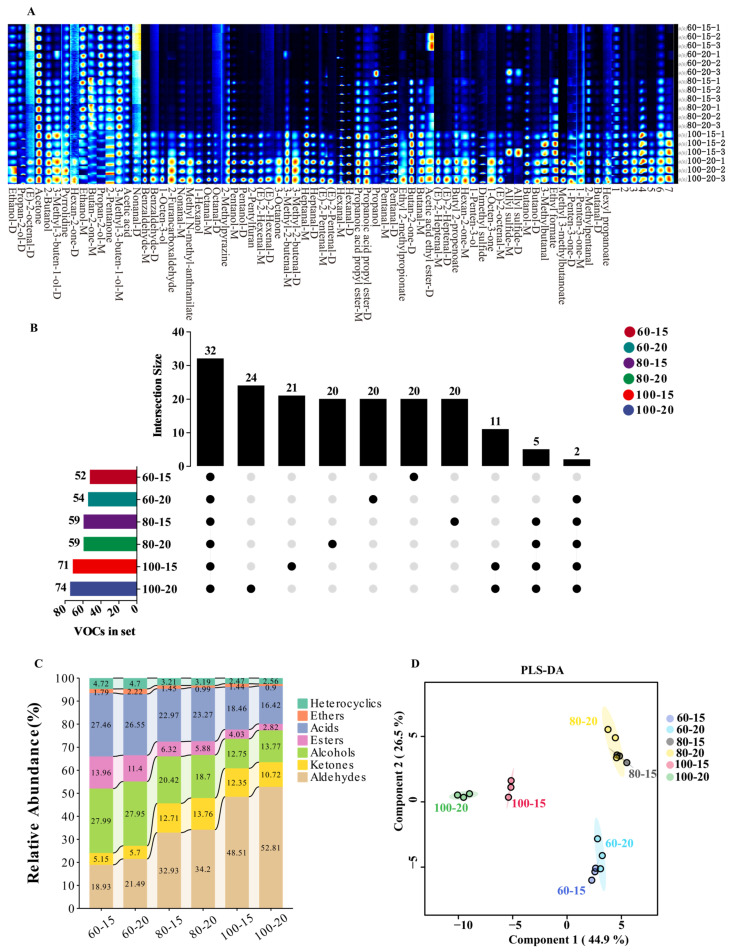
Differences in VOCs incubation conditions. (**A**) Fingerprints profiles of different incubation conditions. (**B**) Upset diagram of VOCs. (**C**) Relative abundance of VOC categories. (**D**) PLS-DA score plot.

**Figure 2 foods-14-04164-f002:**
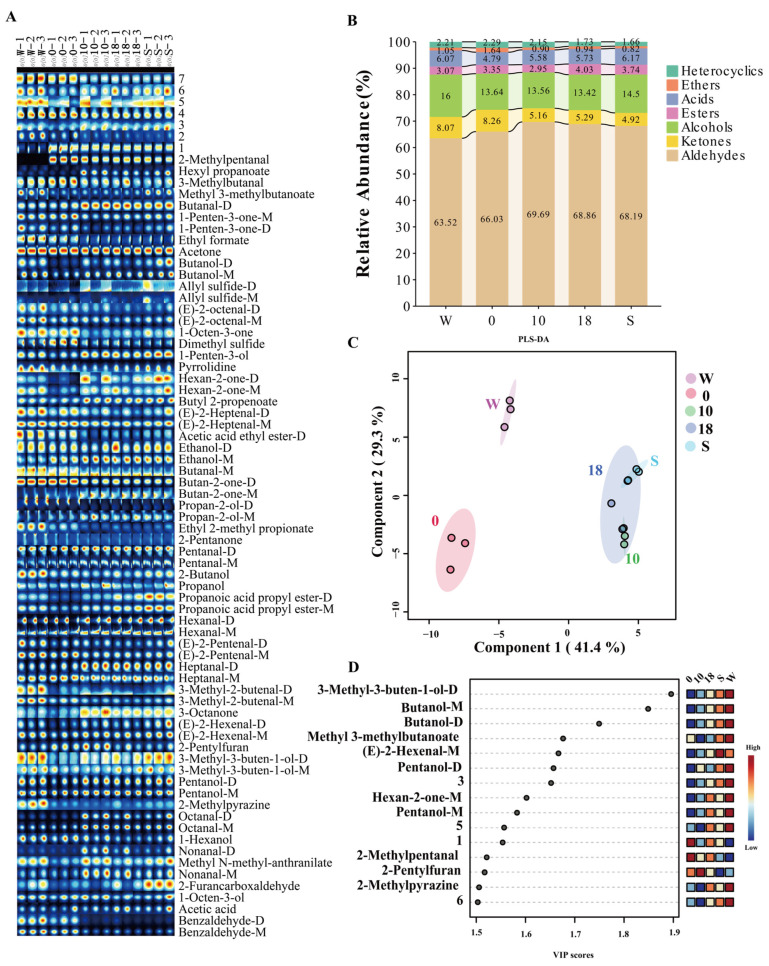
Differences in VOCs medium conditions. W: air (water free), 0: purified water, 10: 10% NaCl, 18: 18% NaCl, S: saturated NaCl (26%, *w*/*v*). (**A**) Fingerprints profiles of different medium conditions. (**B**) Relative abundance of VOC categories. (**C**) PLS-DA score plot. (**D**) VIP score plot.

**Figure 3 foods-14-04164-f003:**
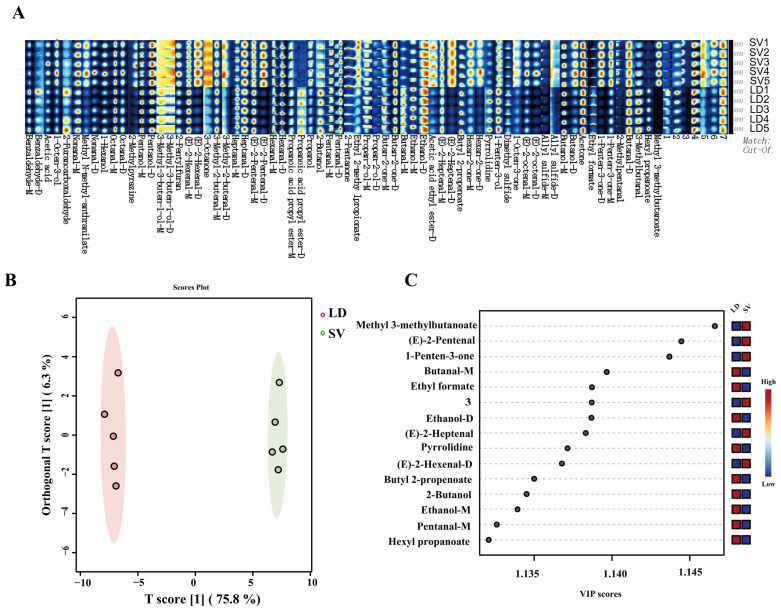
Comparative Analysis of VOCs from longissimus dorsi muscle (LD group) and serratus ventralis muscle (SV group). (**A**) Fingerprints profiles of SV and LD. (**B**) OPLS-DA score plot. (**C**) VIP score plot.

**Table 1 foods-14-04164-t001:** Comparative Analysis of VIP Scores, *p* Values, and ROAV across Different Muscle.

Label	CAS#	VIP Value	*p* Value	*Serratus ventralis*					*Longissimus dorsi*				
				SV1	SV 2	SV3	SV4	SV5	LD1	LD2	LD3	LD4	LD5
(E)-2-octenal-D	C2548870	1.09	<0.01	5.34	5.59	5.17	8.47	6.08	1.52	1.29	1.62	1.59	1.82
1-Octen-3-one	C4312996	1.04	<0.01	151.06	152.40	164.96	164.51	161.07	91.72	77.59	80.47	88.87	76.05
3-Methylbutanal	C590863	1.12	<0.01	27.81	27.43	27.87	29.14	29.34	36.78	33.93	30.48	31.97	32.05
Butanal-D	C123728	1.08	<0.01	30.11	30.96	32.26	29.56	32.90	29.92	27.87	27.14	27.35	27.31
Butanal-M	C123728	1.14	<0.01	2.10	2.24	2.55	1.60	2.31	10.03	9.87	8.34	8.26	7.83
Butyl 2-propenoate	C141322	1.14	<0.01	1.38	1.42	1.57	1.35	1.52	2.13	1.92	1.76	1.75	1.78
Dimethyl sulfide	C75183	1.12	<0.01	16.18	15.02	14.33	13.72	14.00	24.69	21.35	18.82	20.55	19.57
Ethanol-D	C64175	1.14	<0.01	6.31	6.73	6.97	6.09	6.86	8.46	8.43	7.58	8.05	7.65
Heptanal-D	C111717	1.10	<0.01	41.79	43.23	45.23	41.24	44.08	40.77	40.77	40.24	40.38	39.29
Heptanal-M	C111717	1.12	<0.01	12.03	12.03	13.36	10.25	12.52	18.36	17.91	15.62	16.00	14.83
Hexanal-M	C66251	1.13	<0.01	47.71	47.74	51.45	46.19	50.13	53.99	51.09	49.46	49.55	49.69
Hexyl propanoate	C2445763	1.13	<0.01	1.35	1.40	1.36	1.45	1.44	3.03	2.52	2.76	2.86	3.52
Octanal-M	C124130	1.11	<0.01	100.00	100.00	100.00	100.00	100.00	100.00	100.00	100.00	100.00	100.00
Pentanal-D	C110623	1.12	<0.01	13.87	14.19	14.70	13.75	14.70	15.48	14.42	13.49	13.52	13.47
Pentanal-M	C110623	1.13	<0.01	7.80	8.02	8.39	6.96	8.08	9.84	9.32	8.79	8.81	8.79

## Data Availability

The data presented in this study are contained within the article.

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
