# Peer review of "Optimizing GC-IMS for Pork Volatile Fingerprinting: Effects of Incubation Conditions and Medium on Aroma Profiles"

_foods, 2025, doi:10.3390/foods14234164_

Round 1

Reviewer 1 Report

Comments and Suggestions for Authors

The manuscript entitled “Optimizing GC-IMS for Pork Volatile Fingerprinting: Effects of Incubation Conditions and Medium on Aroma Profiles” presents a systematic study on the optimisation of GC–IMS conditions for pork volatile analysis. The topic is relevant to food flavour chemistry and instrumental analysis, and the methodology is competently described. The findings—particularly the identification of optimal incubation conditions (100 °C, 20 min, 10% NaCl medium)—have potential practical value for standardising GC–IMS protocols in meat flavouromics.

However, several aspects of the manuscript require language polishing, clarification of experimental design, and improvement of data interpretation before publication. Specific comments by line number are listed below.

Major Comments

Abstract clarity (Lines 11–27)

The abstract is informative but overly dense. Please consider condensing sentences on statistical methods (lines 17–19) and providing one concise statement of key findings.

Suggest rephrasing “Several statistical methods were used…” (line 17) to “Statistical analyses including t-tests, PLS-DA, and OPLS-DA were employed to assess VOC differences.”

Introduction logic and grammar (Lines 31–78)

At line 35, the phrase “where aroma mainly VOCs” is grammatically incorrect. Suggest “where aroma mainly arises from VOCs”.

Lines 43–46: Avoid over-citation in a single sentence; citations [4, 5] may suffice.

Lines 70–78: The rationale and research gap should be stated more directly. For example, start line 70 with “Despite the increasing use of GC–IMS for meat analysis, no consensus exists on standardised detection parameters.”

Experimental Design clarity (Lines 94–108)

The section heading “2.2. Experimental Design” (line 94) introduces two optimisation experiments. However, the description lacks clarity on replication. Please specify how many biological replicates (animals) and technical replicates (measurements per sample) were performed for each condition.

Line 106: State whether each replicate represents an independent measurement or a repeat of the same sample.

Instrumental Conditions (Lines 117–130)

Line 127: The description “ionization source: tritium source (³H)” should include safety measures and regulatory compliance information.

Line 121: Clarify whether 85 °C needle temperature was empirically optimised or fixed according to instrument recommendations.

Statistical Analysis (Lines 131–154)

The formula at line 150 is not clearly formatted. Use a proper equation format in the revised manuscript.

Clarify the source of thresholds used in ROAV calculations (line 153)—did the authors adapt values from the cited reference [23] or experimentally determine them?

Results – Effect of Incubation Conditions (Lines 155–220)

Line 156: The results are well described, but figures are over-referenced in text. Avoid repeating “as shown in Figure 1A” multiple times.

Lines 173–176: The description of percentage changes should include units (e.g., “relative proportion of aldehydes increased from 18.93% to 52.81%”).

Lines 205–213: The explanation referencing the Arrhenius equation is interesting, but please provide quantitative support (e.g., activation energy or rate constants) if possible.

Line 207: There is a double question mark “reduces?? VOC-binding capacities”—remove.

Results – Effect of Media (Lines 224–266)

Lines 235–240: The mechanistic explanation for moisture reducing oxygen solubility is correct but speculative. Please add a citation supporting this claim.

Line 244: The statement “similar to the study by Zhang et al.” is appropriate but requires specifying the comparable NaCl concentration and species.

Lines 254–260: The discussion could be enriched by mentioning the possible role of ionic strength and protein–flavour interactions in volatile release.

Results – GC–IMS Analysis of Muscles (Lines 270–304)

Lines 282–283: Provide clearer justification for selecting LD and SV muscles beyond fibre-type composition; indicate relevance to consumer flavour differences.

Lines 284–286: Clarify whether “15 key differential compounds” were consistently found across all pigs or averaged from multiple replicates.

Lines 298–302: The explanation linking 1-octen-3-one to fat content is plausible but could be strengthened with quantitative lipid data from the same samples.

Conclusion (Lines 311–320)

The conclusion accurately summarises findings but should emphasise the novelty—i.e., that this is the first systematic optimisation of GC–IMS parameters for pork aroma analysis.

Consider merging lines 317–319 for conciseness.

Minor Comments

Formatting: Ensure consistent use of en dashes (–) for ranges (e.g., 60–100 °C).

References: Check for missing spaces in DOIs and journal titles (e.g., lines 339–340).

Figures: Captions (e.g., lines 221–223 and 267–269) should explain all abbreviations such as “W”, “0”, “S” clearly.

Language: Several minor grammatical issues throughout (e.g., missing articles and plural forms). A thorough language edit by a native English speaker is strongly recommended.

Line 89–93: “long-term storage at −80°C” should read “stored long-term at −80 °C.”

Recommendation

Major Revision

The study is well designed and presents meaningful results for flavouromics research, but revisions are needed to improve clarity, ensure methodological transparency, and enhance scientific precision. Once these issues—particularly those regarding replication details, equation formatting, and language consistency—are addressed.

Comments on the Quality of English Language

The manuscript is generally understandable, but the quality of English requires improvement before publication. Numerous minor grammatical, syntactic, and stylistic errors are present, particularly involving article use, verb tense consistency, and sentence structure. Some phrases are awkward or overly literal translations from Chinese (e.g., lines 35, 205, 207). Scientific terminology is appropriate but occasionally repetitive or imprecise.

A careful professional language edit by a native or proficient English speaker is recommended to ensure fluency, clarity, and conciseness. Attention should also be given to uniform formatting of units, en dashes, and reference styles throughout the text.

Author Response

Dear reviewers,

Thank you and all the very much for your valuable comments on our manuscript submitted to Foods (Manuscript ID: 3973213). We greatly appreciate the opportunity to further revise our manuscript. We have carefully checked the manuscript and revised it according to these comments, and all the changes have been marked in the revised manuscript. We submit here the revised manuscript as well as a list of changes.

Comment 1: Abstract clarity (Lines 11–27)

The abstract is informative but overly dense. Please consider condensing sentences on statistical methods (lines 17–19) and providing one concise statement of key findings.

Suggest rephrasing “Several statistical methods were used…” (line 17) to “Statistical analyses including t-tests, PLS-DA, and OPLS-DA were employed to assess VOC differences.”

Response 1 Thank you very much for your rigorous academic attitude.

The comment regarding the description of the statistical methods has been carefully considered, and the text in the abstract has been amended accordingly (please see page 1, lines 16-17).

Comment 2: Introduction logic and grammar (Lines 31–78)

2.1. At line 35, the phrase “where aroma mainly VOCs” is grammatically incorrect. Suggest “where aroma mainly arises from VOCs”.

2.2. Lines 43–46: Avoid over-citation in a single sentence; citations [4, 5] may suffice.

2.3. Lines 70–78: The rationale and research gap should be stated more directly. For example, start line 70 with “Despite the increasing use of GC–IMS for meat analysis, no consensus exists on standardised detection parameters.”

Response 2Thanks for your valuable comments.

2.1. The grammatical error at the indicated location has been corrected (please see page 1, lines 32-33).

2.2. We have streamlined the reference citations to a more appropriate number (please see pages 2-3, lines 43-46).

2.3. The opening sentence has been rephrased and the paragraph has been restructured to improve clarity and flow (please see page 3, lines 69-72).

Comment 3: Experimental Design clarity (Lines 94–108)

The section heading “2.2. Experimental Design” (line 94) introduces two optimisation experiments. However, the description lacks clarity on replication. Please specify how many biological replicates (animals) and technical replicates (measurements per sample) were performed for each condition.

Line 106: State whether each replicate represents an independent measurement or a repeat of the same sample.

Response 3We are grateful to the reviewer for the comment regarding experimental rigor.

We have added text to explicitly state the number of experimental replicates (n=3) and to clarify that each replicate constitutes an independent measurement, thereby providing greater clarity on the sample size and the nature of replicates (please see page 3, lines 94-96, 98-100).

Comment 4: Instrumental Conditions (Lines 117–130)

4.1. Line 127: The description “ionization source: tritium source (³H)” should include safety measures and regulatory compliance information.

4.2. Line 121: Clarify whether 85 °C needle temperature was empirically optimised or fixed according to instrument recommendations.

Response 4Thanks for your kind reminding.

4.1. We have added the activity level of the ionization source. This activity falls below the 1 GBq threshold defined as an exempt source by the Chinese National Standard GB 18871-2002, thereby confirming that our operations are in full compliance with all relevant safety regulations (please see page 4, lines 136-138).

4.2. The temperature setting for the injection needle has now been added to the manuscript. It was maintained at the instrument's default configuration as this is the established standard for this type of analysis (please see see page 4, line 130).

Comment 5: Statistical Analysis (Lines 131–154)

5.1. The formula at line 150 is not clearly formatted. Use a proper equation format in the revised manuscript.

5.2. Clarify the source of thresholds used in ROAV calculations (line 153)—did the authors adapt values from the cited reference [23] or experimentally determine them?

Response 5Thanks for your detailed comments.

5.1. The equation has been revised to include a more detailed description of its individual parameters to enhance clarity (please see page 5, lines 169-173).

5.2. The thresholds used for calculating the ROAV values were based on the established methodology from reference Compilations of Flavour Threshold Values in Water and Other Media[26], as described in the manuscript (please see page 5, line 173).

Comment 6: Results – Effect of Incubation Conditions (Lines 155–220)

6.1. Line 156: The results are well described, but figures are over-referenced in text. Avoid repeating “as shown in Figure 1A” multiple times.

6.2. Lines 173–176: The description of percentage changes should include units (e.g., “relative proportion of aldehydes increased from 18.93% to 52.81%”).

6.3. Lines 205–213: The explanation referencing the Arrhenius equation is interesting, but please provide quantitative support (e.g., activation energy or rate constants) if possible.

6.4. Line 207: There is a double question mark “reduces?? VOC-binding capacities”—remove.

Response 6Thanks for your detailed comments.

6.1. Phrases such as "as shown in Figure 1A" have been revised throughout the text to improve the academic phrasing, with changes reflected in multiple locations (e.g., lines 176, 180, 184).

6.2. The description of the percentage changes has been revised accordingly (please see page 6, lines 193-196).

6.3. The Arrhenius equation describes the exponential dependence of the reaction rate constant on temperature. The release of VOCs has been reported to be highly sensitive to temperature in the literature, a detailed mechanistic investigation of this relationship is beyond the scope of our present study. Exploring this underlying mechanism represents a valuable direction for our future work.

6.4. All extraneous "?" characters have been corrected in the manuscript text.

Comment 7: Results – Effect of Media (Lines 224–266)

7.1 Lines 235–240: The mechanistic explanation for moisture reducing oxygen solubility is correct but speculative. Please add a citation supporting this claim.

7.2 Line 244: The statement “similar to the study by Zhang et al.” is appropriate but requires specifying the comparable NaCl concentration and species.

7.3 Lines 254–260: The discussion could be enriched by mentioning the possible role of ionic strength and protein–flavour interactions in volatile release.

Response 7Thanks for your kind reminding.

7.1. We have now cited References at the indicated location (please see page 8, lines 255-257), which provide a detailed discussion of the action mechanism.

7.2. We have added relevant supporting references and expanded the discussion on the brine concentration comparison (please see page 8, lines 264-273).

7.3. We have enhanced the discussion on the interaction between saline and protein-flavor compounds accordingly (please see page 8, lines 285-290).

Comment 8: Results – GC–IMS Analysis of Muscles (Lines 270–304)

8.1 Lines 282–283: Provide clearer justification for selecting LD and SV muscles beyond fibre-type composition; indicate relevance to consumer flavour differences.

8.2 Lines 284–286: Clarify whether “15 key differential compounds” were consistently found across all pigs or averaged from multiple replicates.

8.3 Lines 298–302: The explanation linking 1-octen-3-one to fat content is plausible but could be strengthened with quantitative lipid data from the same samples.

Response 8Thanks for your valuable guidance.

8.1. We have included the justification for utilizing LD and SV, while also consolidating the various factors that impact consumer sensory assessment, thereby providing clearer context for our methodological choices (please see page 10, lines 303-313).

8.2. We have incorporated a statement clarifying that the 15 key differential compounds were consistently detected across all samples, underscoring their reliability (please see page 10, lines 325-326).

8.3. We have included relevant references that provide evidential support for the observed difference in IMF content between SV and LD (please see page 10, lines 307-308).

Comment 9: Conclusion (Lines 311–320)

The conclusion accurately summarises findings but should emphasise the novelty—i.e., that this is the first systematic optimisation of GC–IMS parameters for pork aroma analysis.

Consider merging lines 317–319 for conciseness.

Response 9Thanks for your comments.

We have added text to better highlight the novelty of our study and have rewritten the conclusion for greater clarity (please see page 13, lines 356-370).

Comment 10: Minor Comments

10.1. Formatting: Ensure consistent use of en dashes (–) for ranges (e.g., 60–100 °C).

10.2. References: Check for missing spaces in DOIs and journal titles (e.g., lines 339–340).

10.3. Figures: Captions (e.g., lines 221–223 and 267–269) should explain all abbreviations such as “W”, “0”, “S” clearly.

10.4. Language: Several minor grammatical issues throughout (e.g., missing articles and plural forms). A thorough language edit by a native English speaker is strongly recommended.

10.5. Line 89–93: “long-term storage at −80°C” should read “stored long-term at −80 °C.”

Response 10Thanks for your valuable comments.

10.1. The use of en dashes has now been standardized throughout the manuscript to ensure consistent adherence to formatting conventions.

10.2. The formatting of all references has been verified and standardized throughout the manuscript to ensure consistency with the journal's style.

10.3. We have checked and corrected the formatting of the references throughout the manuscript.

10.4. As suggested, we have checked and corrected non-standard expressions in the text to enhance readability (e.g., line 107, line 166).

10.5. We have rephrased the text as suggested.

Thank you and all the reviewers again for the kind advices. If you have any questions about this revision, please don’t hesitate to let me know.

Reviewer 2 Report

Comments and Suggestions for Authors

Article

Optimizing GC-IMS for Pork Volatile Fingerprinting: Effects of Incubation Conditions and Medium on Aroma Profiles

Lei Yu, Binbin Wang, Ziwei Xu, Kaili Ge, Yihan Yuan, Xiangbin Ding, Xiaoming Men*, Keke Qi

        In an interesting and well-designed study, the authors used GC-IMS to determine the effects of incubation (temperature/duration) and environment (water and different concentrations of NaCl) on volatile organic compounds (VOCs) in pork. Quantification of differences in VOCs was determined using several statistical methods [t-test, partial least squares analysis with discriminant (PLS-DA), orthogonal partial least squares analysis with discriminant (OPLS-DA), and relative odor activity value (ROAV)]. The entire study is presented very precisely, clearly and legibly, without redundant descriptions. The size of Article is highly desirable (12 pages in total, with References). The priority is given to the interpretation of the results and their discussion, so it is the most extensive part of the Article. The conclusions are correctly formulated; they can be improved by thinking about deepening the already realized researches in the future.

        The Abstract is in accordance with the Instructions of the authors. The Abstract contain the briefest results of the research.

        Introduction chapter: the introductory chapter is informative enough, the reader can understand what the "state-of-art" is about the detection of volatile organic compounds (VOCs) with a modern method such as gas chromatography-ion mobility spectrometry (GC-IMS), and the fact that the key influencing factors in GC-IMS analysis are incubation temperature, incubation time, and sampling medium is emphasized. Also, at the end of the Introduction chapter, the goal and tasks of the examination are well defined. Sixteen references are listed in the Introduction.

        Material and methods chapter has 4 subchapters. The way of describing the design of the experiment and the applied methods is very clear, precise and enables the reproducibility of the experiment. The description of the statistical tools and methods used is very precise.            

        The Results One Table and 3 Figures (with multiple included views each) reveal extremely interesting results, illustrative, engaging and not confusing the reader. The graphics dominate their quality, and are very clear and illustrative. Figures 1 and 2 have 4 segments each (A-D), Figure 3 have 3 segments (A-C), thematically connected. The results are very useful because they lead to the development of flavoromics, which has a strong impact in the era of dominance of sensory attributes of meat and meat products, which attract consumers. The results indicate that a GC-IMS experiment under determined (various) conditions can determine how to most effectively identify volatile substances in different pork samples, as well as further determine differences in the types and content of volatile organic compounds in pork.

        The Conclusions: The conclusions chapter are correctly formulated; they can be improved by thinking about deepening the already realized researches in the future.

        References are the appropriate, the number of cited references (45) is optimal for Article. They are recently, references from the last 5 years predominate and a very small number of older but high-quality references, with a few self-citations.

        There are a few technical shortcomings (I put all in the Manuscript, taken from the SuSy platform).

Author Response

Dear reviewers,

Thank you and all the very much for your valuable comments on our manuscript submitted to Foods (Manuscript ID: 3973213). We greatly appreciate the opportunity to further revise our manuscript. We have carefully checked the manuscript and revised it according to these comments, and all the changes have been marked in the revised manuscript. We submit here the revised manuscript as well as a list of changes.

Comment 1: In this place, the symbol for degrees Celsius is correctly entered, with a difference of digits, and the symbol for degrees and C are without spaces. Please make it uniform in all places in the text of the manuscript, because it is currently different.

Response 1Thanks for your detailed comments.

We have checked and corrected the use of symbols throughout the manuscript.

Comment 2: Should the entered question mark symbol be deleted? In any case, it should not be present in that place in the manuscript.

Response 2Thanks for your kind reminding.

We have removed all extraneous "?" characters that were present in the text.

In addition, we have added text to better highlight the novelty of our study and have rewritten the conclusion for greater clarity (please see page 13, lines 356-370).

Thank you and all the reviewers again for the kind advices. If you have any questions about this revision, please don’t hesitate to let me know.

Reviewer 3 Report

Comments and Suggestions for Authors

The manuscript entitled “Optimizing GC-IMS for Pork Volatile Fingerprinting: Effects of Incubation Conditions and Medium on Aroma Profiles” addresses an interesting topic related to aroma compounds and enviorenmental factors on meat products. However, several aspects in the Introduction, Materials and Methods, and Results and Discussion sections require clarification to enhance the transparency and clarity of the manuscript.

The manuscript will be considered after fullfilment of following comments:

In introduction, the authors should explain what other techniques are available for the detection of volatile components.

The introduction  is very lack of the studies to comparison with the literature what has been worked on this topic. The authors should sthrengthen this part.

Line 104: The authors stated that “five media were evaluated to identify the optimal medium: pure water, 10% NaCl solution, 18% NaCl solution, and saturated NaCl solution (26%)…”. However, only four media are listed, not five. This inconsistency should be checked and corrected.

Line 117: The compounds mentioned are alkane mixtures, not internal standards. Internal standards should be included in each sample during every GC-IMS run to ensure reliable quantification.

Line 135: The authors should provide the calibration curves along with all relevant statistical results of the developed models to support the validity of their findings.

Figures 1A and 2A: Both figures should be improved to enhance clarity and readability. The VOC labels are difficult to read, and the results cannot be clearly distinguished among the different samples. The authors are advised to enlarge or reorganize the graphical elements to make the figures more understandable.

Line 245: The effect of the medium on different types of meat has not been adequately discussed. The authors should provide a brief literature comparison addressing how media influence has been reported for other meat types.

Additionally, the rationale for selecting only a 10% NaCl brine solution should be clarified. Why was only one concentration tested? It would be more informative to evaluate the effect of different brine concentrations (e.g., 5%, 10%, and 15% or 3%, 5%, and 10%) to better understand the impact of salinity. The authors are encouraged to make an analyze within three points and discuss the results using multiple concentration points.

The conclusion section should be expanded to include more detailed information, particularly the statistical results of the models. In addition, it should be rewritten to highlight the main findings and provide suggestions for further investigations in this research area.

Please check and correct punctuation and formatting errors throughout the manuscript to ensure clarity and consistency (some of the mistakes are listed below):

L 207, denaturation reduces??  

L201, (e.g., furans, pyrazines)[24]. (there are many mistakes such as throughout the manuscript)         

L 210,  60°C and 210 80°C??)

Author Response

Dear reviewers,

Thank you and all the very much for your valuable comments on our manuscript submitted to Foods (Manuscript ID: 3973213). We greatly appreciate the opportunity to further revise our manuscript. We have carefully checked the manuscript and revised it according to these comments, and all the changes have been marked in the revised manuscript. We submit here the revised manuscript as well as a list of changes.

Comment 1: In introduction, the authors should explain what other techniques are available for the detection of volatile components.

Response 1Thanks for your valuable guidance.

We have expanded the discussion to include other techniques available for the detection of volatile compounds (please see lines 36-39).

Comment 2: The introduction is very lack of the studies to comparison with the literature what has been worked on this topic. The authors should sthrengthen this part.

Response 2Thanks for your valuable guidance.

We have incorporated relevant comparative studies to better contextualize our research (please see lines 63-70).

Comment 3: Line 104: The authors stated that “five media were evaluated to identify the optimal medium: pure water, 10% NaCl solution, 18% NaCl solution, and saturated NaCl solution (26%)…”. However, only four media are listed, not five. This inconsistency should be checked and corrected.

Response 3Thanks for your comments.

We have reviewed the wording in this section and corrected the identified errors (please see line 105).

Comment 4: Line 117: The compounds mentioned are alkane mixtures, not internal standards. Internal standards should be included in each sample during every GC-IMS run to ensure reliable quantification.

Response  4Thank you very much for your rigorous academic attitude.

We fully agree with this insightful comment. In the present study, we employed signal intensity-based relative quantification to assess sample repeatability and thereby demonstrate the feasibility of our methodology. We acknowledge that the inclusion of an internal standard would provide more robust quantification. Therefore, in subsequent experiments, we will incorporate internal standards to ensure greater reliability of the quantitative results.

Comment 5: Line 135: The authors should provide the calibration curves along with all relevant statistical results of the developed models to support the validity of their findings.

Response 5Thanks for your valuable comments.

We have added the formula used to calculate the compound Retention Indices (RI) from Retention Time (RT) data in the manuscript. (please see page 5, lines 130-135).

Comment 6: Figures 1A and 2A: Both figures should be improved to enhance clarity and readability. The VOC labels are difficult to read, and the results cannot be clearly distinguished among the different samples. The authors are advised to enlarge or reorganize the graphical elements to make the figures more understandable.

Response 6Thanks for your comment.

The clarity of the figures has been enhanced to ensure that the VOC labels are now clearly visible.

Comment 7: Line 245: The effect of the medium on different types of meat has not been adequately discussed. The authors should provide a brief literature comparison addressing how media influence has been reported for other meat types.

Additionally, the rationale for selecting only a 10% NaCl brine solution should be clarified. Why was only one concentration tested? It would be more informative to evaluate the effect of different brine concentrations (e.g., 5%, 10%, and 15% or 3%, 5%, and 10%) to better understand the impact of salinity. The authors are encouraged to make an analyze within three points and discuss the results using multiple concentration points.

Response 7Thanks for your valuable comments.

We have added supporting references and expanded the discussion regarding the saline concentration comparison (please see lines 232-240).

Comment 8: The conclusion section should be expanded to include more detailed information, particularly the statistical results of the models. In addition, it should be rewritten to highlight the main findings and provide suggestions for further investigations in this research area.

Response 8Thanks for your comment.

We have revised the conclusion by incorporating key findings from the data analysis and have added perspectives for future research directions (please see page 13, lines 315-327).

Comment 9: Please check and correct punctuation and formatting errors throughout the manuscript to ensure clarity and consistency (some of the mistakes are listed below):

L 207, denaturation reduces?? 

L201, (e.g., furans, pyrazines)[24]. (there are many mistakes such as throughout the manuscript)        

L 210,  60°C and 210 80°C??)

Response 9Thanks for your detailed comments.

We have corrected the errors noted above and performed a thorough check of the punctuation throughout the manuscript.

Thank you and all the reviewers again for the kind advices. If you have any questions about this revision, please don’t hesitate to let me know.

Reviewer 4 Report

Comments and Suggestions for Authors

The manuscript is interesting, innovative and provides very valuable new information on the optimization of GC-IMS for the determination of volatile compounds in pork. The influence of incubation conditions and the content of sodium chloride in aqueous solutions on the aroma profiles was studied.

The article offers a thorough reading of the data and is distinguished by balanced results in quantity and quality, supported by solid statistical processing. The results are presented attractively and at the same time accessible and informative. The methods of analysis used are distinguished by completeness and scope. The article contributed to the development of analytical chemistry in identifying an optimized analytical procedure for the determination of volatile organic compounds during thermal processing of pork. The results obtained are skillfully interpreted and explained with appropriate discussion and fundamental scientific arguments. A sufficient number of new scientific articles are used in the introduction and discussion. This further increases the scientific added value of the manuscript. The following conclusions directly follow from the obtained results and justify optimal conditions for the analysis of VOCs in pork using GC-IMS, namely - addition of 10% NaCl solution and incubation at 100°C for 20 minutes. Under these incubation conditions, the highest number and content of VOCs with relatively stable repeatability was established.

Author Response

Dear reviewers,

Thank you and all the very much for your valuable comments on our manuscript submitted to Foods (Manuscript ID: 3973213). We greatly appreciate the opportunity to further revise our manuscript. We have carefully checked the manuscript and revised it according to these comments, and all the changes have been marked in the revised manuscript. We submit here the revised manuscript as well as a list of changes.

Comment :

  1. How many pigs were slaughtered in this experiment?
  2. How (under what conditions) and for how long was the carcass meat cooled?
  3. What was the final temperature (24 h after death) in the center of the thigh muscles?
  4. How many days after death were samples taken from the longissimus dorsi muscle? This question is related to the period of postmortem changes in which the meat was studied.
  5. Was it in the phase of postmortem rigor mortis or of initial autolysis - maturation?
  6. How many repetitions (n) were made?!

ResponseThanks for your valuable guidance, we would provided more detailed information (please see page 2, lines 84-85, lines 87-90).

  1. In this study, a total of six pigs were slaughtered. Tissue samples from one animal were used for method optimization, while samples from the remaining five were used to validate the optimized conditions.
  2. The carcasses were stored at 4°C for 24 hours to allow for the complete resolution of rigor mortis.
  3. The final temperature in the center of the thigh muscles, measured 24 hours post-mortem, had equilibrated to the ambient temperature of approximately 4°C
  4. According to the standardized protocol, sampling was conducted at 24 hours post-mortem.
  5. Our sampling protocol strictly maintained the carcasses in the initial autolysis to maturation stage to ensure sample consistency and data reliability
  6. In the method optimization phase of this study, each independent sample was analyzed once, with the entire experiment repeated in three technical replicates. For the validation phase under optimal conditions, the experiment was conducted with five biological replicates

Thank you and all the reviewers again for the kind advices. If you have any questions about this revision, please don’t hesitate to let me know.

Round 2

Reviewer 3 Report

Comments and Suggestions for Authors

There are still several major deficiencies in the manuscript:

“The authors did not correctly understand Comment 4. The compounds mentioned are alkane mixtures used for retention index calibration, not internal standards.

The sentence ‘For the precise calculation of the retention indices (RI) of VOCs, six methyl ketones were used as internal standards: 2-butanone, 2-pentanone, 2-hexanone, 2-heptanone, 2-octanone, and 2-nonanone’ should be corrected to ‘reference standards for retention index calculation’ or ‘Kovats index reference compounds’ instead of ‘internal standards’.

The authors should be more careful regarding the terminology used for chemical substances.”

In Comment 5, the issue is that the calibration curves referenced by the authors were not provided anywhere in the manuscript—neither in the main text nor in the Supporting Information. Calibration curves are essential for validating quantitative analyses. The authors should provide all calibration curves (with equations, R² values, and concentration ranges) either in the main text or as part of the Supporting Information.”

The line numbers in the manuscript do not match the ones provided in the Author’s Response Letter.

Author Response

Dear reviewer,

Thank you very much once again for your valuable comments and suggestions on our manuscript (ID: 3973213) submitted to Foods. We sincerely appreciate the opportunity to further revise our manuscript based on your additional feedback. We have carefully reviewed all the comments and made corresponding revisions, which have been highlighted in the updated manuscript. Attached please find the revised version of our manuscript, along with a point-by-point response to the comments

Comment 1: The authors did not correctly understand Comment 4. The compounds mentioned are alkane mixtures used for retention index calibration, not internal standards.

The sentence ‘For the precise calculation of the retention indices (RI) of VOCs, six methyl ketones were used as internal standards: 2-butanone, 2-pentanone, 2-hexanone, 2-heptanone, 2-octanone, and 2-nonanone’ should be corrected to ‘reference standards for retention index calculation’ or ‘Kovats index reference compounds’ instead of ‘internal standards’.

The authors should be more careful regarding the terminology used for chemical substances.”

Response 1We sincerely thank the reviewer for their continued attention and thoughtful guidance. We have corrected the inaccurate description concerning the use of chemical materials in the manuscript (please see page 4, lines 125-127).

Comment 2: In Comment 5, the issue is that the calibration curves referenced by the authors were not provided anywhere in the manuscript—neither in the main text nor in the Supporting Information. Calibration curves are essential for validating quantitative analyses. The authors should provide all calibration curves (with equations, R² values, and concentration ranges) either in the main text or as part of the Supporting Information.”

Response 2We are grateful to the reviewer for the guidance on the calibration curve. We have revised the manuscript to include the specific formula, the coefficient of determination (R²), and the concentration range (please see page4, line 146 and page 5, lines 148-150).

Comment 3: The line numbers in the manuscript do not match the ones provided in the Author’s Response Letter.

Response 3We appreciate the reviewer's attention to this detail. The line numbering has been checked and updated throughout the manuscript.

Thank you again for the kind advices. If you have any questions about this revision, please don’t hesitate to let me know.